# The Impact of Circular Exercise Diameter on Bone and Joint Health of Juvenile Animals

**DOI:** 10.3390/ani12111379

**Published:** 2022-05-27

**Authors:** Alyssa A. Logan, Brian D. Nielsen, Kristina M. Hiney, Cara I. Robison, Jane M. Manfredi, Daniel D. Buskirk, John M. Popovich

**Affiliations:** 1Department of Animal Science, Michigan State University, 474 S. Shaw Ln., East Lansing, MI 48824, USA; bdn@msu.edu (B.D.N.); oconn107@msu.edu (C.I.R.); buskirk@msu.edu (D.D.B.); 2Department of Animal and Food Sciences, Oklahoma State University, 201J Animal Sciences, Stillwater, OK 74074, USA; khiney@okstate.edu; 3Department of Pathobiology and Diagnostic Investigation, Michigan State University, 784 Wilson Rd., East Lansing, MI 48824, USA; manfred1@msu.edu; 4Center for Neuromusculoskeletal Clinical Research, Department of Osteopathic Manipulative Medicine, Michigan State University, 965 Wilson Rd., B439, East Lansing, MI 48824, USA; popovi16@msu.edu

**Keywords:** lunging, walker, horse, equine, bone, skeleton, joint, osteoarthritis, calf, gait

## Abstract

**Simple Summary:**

In many equestrian disciplines, circular exercise is utilized to train, exercise, and compete with horses and can vary in gait, as well as diameter. This study aimed to determine if circular exercise diameter impacts animal health. Calves have previously been used as a terminal skeletal model of juvenile horses, allowing for the collection of musculoskeletal samples that are not acceptable from horses. Calves exercised on a small circle (12-m clockwise), large circle (18-m clockwise), treadmill, or served as non-exercised controls. Exercise was performed at a walking speed, starting at 5 min per day and increasing 5 min weekly until reaching 30 min per day during the 7-week study. The response to exercise was monitored in the forelimb bones and joints. The small circular exercise group was found to have bone diameters that differed between the right and left fused third and fourth metacarpi, and between lateral and medial proximal phalanx bones. Cartilage glycosaminoglycan content was greater in the outside leg of the small circle exercise calves than the inside leg, with no differences noted within other treatments. These differences suggest that altering circular exercise diameter can impact bone and joint health, and that larger diameter circles may prevent asymmetric loading between inside and outside legs.

**Abstract:**

Circular exercise is used in many equestrian disciplines and this study aimed to determine if circle diameter impacts juvenile animal forelimb bone and joint health. On day 0, 24 calves at 9 weeks of age were assigned the following exercise treatments: small circle (12 m clockwise), large circle (18-m clockwise), treadmill, or non-exercised control. Exercise was initiated at 1.1–1.5 m/s for 5 min/d and increased 5 min weekly until reaching 30 min/d. On day 49, synovial fluid was collected from multiple joints, cartilage was collected from the proximal surface of fused third and fourth metacarpi (MC III and IV), and forelimbs underwent computed tomography scans. A statistical analysis (PROC mixed) was performed in SAS 9.4. The inside leg of the small circle treatment had a larger MC III and IV dorsopalmar external diameter than the outside (*p* = 0.05). The medial proximal phalanx had a greater mediolateral diameter than the lateral proximal phalanx of the small circle treatment (*p* = 0.01). Fetlock nitric oxide was greater in the large circle and treadmill treatments (*p* < 0.0001). Cartilage glycosaminoglycan concentration was greater in the outside leg of the small circle exercise treatment than the inside leg (*p* = 0.03). Even at slow speeds, circular exercise diameter can impact joint and bone health, but faster speeds may have greater alterations.

## 1. Introduction

The strain environment of bone can be influenced by exercise and consists of many factors, such as magnitude of strain, rate of strain, distribution of strain, and frequency of strain. The response of bone to exercise can be explained by the mechanostat theory. Under this theory, bone can adapt to its mechanical environment based on the strains it is subject to. When there is too little strain in the environment, the amount of bone needed is reduced, and bone resorption occurs. When there is an increase in strain, the amount of bone needed is increased, and bone formation occurs. The mechanostat theory also predicts that bone can be maintained and repaired without net resorption or net formation when strain is high enough to prevent the removal of unnecessary bone, but not too high to elicit an acquisition of more bone [1]. Dynamic exercise, such as sprinting, leads to an increase in bone strength in juvenile animals [2]. While straight-line sprint exercise has been found to benefit bone strength, little is known of the impacts that circular exercise has on bone or joint health of juvenile animals. Circular exercise is used frequently in training, exercise, and competition of horses across various disciplines. Common methods of circular exercise include riding under saddle, or working on the ground, such as in round pens and lunging. In the early training of a young horse, circular exercise via a round pen or lunging is used frequently. During circular exercise, the gait (walk, trot, canter, or gallop) and diameter can be altered by the handler or rider.

While exercising on a circle, horses will lean into the center of the circle by up to 20 degrees to maintain balance [3,4]. Speed and radius have been found to impact lean angle, with greater speed and smaller radii increasing the lean-in towards the center of the circle [5]. The level of training can also impact lean angle, as a horse acclimated to traveling on a circle can travel more upright than a horse that is not acclimated to traveling in a circle or arc. This has especially been noted in dressage, where trained horses are able to travel “on the bit”, while inexperienced horses may not be able to engage the back and hindlimb musculature through a circle [6]. Quadrupeds, such as horses, may be at an advantage compared to bipedal runners during curve running, as they can redistribute weight to multiple stance legs within a stride. It has been found that while cantering in a 10-m circle, horses will have greater peak ground reaction force on the outside forelimb compared to the inside forelimb [7]. While traveling around a curve, Thoroughbred racehorses experience greater strain to the outside forelimb, which increases as speed increases [8]. When both humans and horses sprint through a curve, the outside limb is known to generate more vertical and lateral force than the inside limb [5].

The sharpness of a turn is also found to impact horses and human runners. A sharply curved track (5-m radius) leads to greater torsion on the inside tibia of humans, compared to running on a gently curved track (15-m) or straight line [9]. When running around a curve, the inside and outside limbs of human subjects are not biomechanically symmetrical. As the circle radius decreases, the peak resultant ground reaction forces to the inside leg decrease compared to straight line running [10]. As speed increases, the whole forelimb and third metacarpal angle in the horse increase, supporting the observation that, due to centripetal acceleration while traveling around a turn, horses lean into a turn at a greater angle as speed increases [5]. The area loaded by the outside front hoof is greater at the canter than the trot or walk during circular exercise, suggesting a push-off motion with the outside front leg at the canter [11]. Due to the reduced surface area of the inside hoof that is loaded, uneven forces may be placed on joints and bones of the fore and hind limbs during circular exercise. These forces exerted on a smaller surface area have the potential to lead to a higher risk for joint injury and osteoarthritis [3].

Circle radius can impact animal and rider safety and a retrospective study of risk factors for jockeys in Japanese Thoroughbred racing found smaller tracks to have a greater risk of injury [12]. Osteoarthritis and joint injuries have been reported as a leading cause of lost training days and horse wastage, yet little research has bridged the gap between circular exercise and joint damage. Osteoarthritis can be caused by excessive loads to normal cartilage, or normal loads to abnormal cartilage [13,14]. With up to 60% of lameness being related to osteoarthritis, this gap in research greatly affects the equine community [15,16].

Biomarkers can allow the determination of bone and joint activity throughout an exercise trial. Osteocalcin (OC), a marker of bone formation, and c-telopeptide crosslaps of type I collagen (CTX-1), a marker of bone resorption, can be monitored simultaneously from serum samples to aid in determining bone activity in horses and bovines [2,17]. Procollagen II c-propeptide (CPII) is a marker of collagen synthesis [18]. Osteoarthritis-affected horses have been found to have greater CPII serum and synovial fluid concentrations during an exercise trial than horses without osteoarthritis [19]. Prostaglandin E2 (PGE_2_) and nitric oxide (NO) are markers of inflammation and cartilage metabolism that are frequently evaluated in synovial fluid. Synovial fluid PGE_2_ is found to be greater in joints affected by osteoarthritis than in joints that are not affected by osteoarthritis in exercising horses [19]. Exercise has also been found to increase the synovial fluid concentration of PGE_2_ in healthy, sound horses [20]. Joints effected by osteoarthritis may experience both catabolic and protective functions of NO [21]. Glycosaminoglycans (GAG) are a hydrophilic proteoglycan in the extracellular matrix of articular cartilage, which provide shock absorption and resistance against forces [22]. Within articular cartilage, the GAG concentration had been found to be heterogenous between joint locations [22,23]. Biomarkers can be combined with the evaluation of bone morphology as well as tensile testing to determine the activity, shape, and strength of bone, as well as joint health [2,24].

Many studies have evaluated the gait characteristics of animals performing circular exercise, but few have evaluated bone and joint health. The objective of this study was to utilize a calf model to determine the impact of diameter during circular exercise at the walk to forelimb bone and joint health in juvenile animals. Calves have been used as a model for the management and exercise of young horses successfully in previous studies [2,23,24,25]. It was hypothesized that exercise on a smaller diameter circle would lead to increased biological markers of joint inflammation and metabolism and greater asymmetry between inside and outside forelimbs.

## 2. Materials and Methods

### 2.1. Animals, Housing and Exercise

Holstein steer calves, previously dehorned, were obtained from a local farm at 8 weeks of age and moved into group-housing at the Michigan State University (MSU) Veterinary Research Farm. Group-housing consisted of a pen (112 m^2^), including a partially enclosed three-sided shed (17 m^2^) bedded in straw with a feeding trough and automatic water trough. Calves had ad libitum access to calf starter (Caledonia Farmers Elevator, Caledonia, MI, USA) and water. Calves were given one week for acclimation to housing and halter training before exercise began at 9 weeks of age. Each calf underwent two days of halter training, in which they were introduced to the sensation of a rope halter and taught to move forward from the halter pressure, while being led for two laps in the group-housing pen. Calves were evaluated with daily health checks to assure the safety of exercise.

On day 0, the calves were 9 weeks of age, and blood samples were taken along with height, weight, and length measurements. The calves were then randomly striated to treatment groups based on weight. The calves remained in their treatment groups for 7 weeks (49 d). The treatment groups consisted of non-exercise controls, treadmill exercise, small circle exercise, and large circle exercise. All treatment groups were maintained together in a group-housing pen. The calves randomized to the circle and treadmill exercise treatments exercised 5 d/wk starting at 5 min/d and increased by 5 min each week until reaching 30 min (Table 1). Exercise was performed at a speed of 1.1–1.5 m/s, allowing the calves to maintain a walking gait. This exercise protocol allowed the calves to acclimate to exercise throughout the study. Circular exercise was performed in a clockwise fashion, tracking to the right, on a mechanical walker (Q-line horse exercise), with both a small diameter (12 m) and large diameter (18 m) track. Treadmill exercise was performed on a Classic Treadmill equine treadmill (Classic Champion Model 940, Queensland, Australia).

### 2.2. Sample Collection

Starting on day 0 and continuing weekly, serum was collected via jugular venipuncture into non-heparinized vacutainers. Calf height, weight, and length were recorded weekly, starting on day 0. Height was measured from the floor to the top of the withers, with an L-shaped measuring stick that was adjustable to the wither height of the calf. Weight was measured with a weight scale (Tru-Test; Model 700; Mount Wellington, New Zealand) and length was from the point of shoulder to ischium, or pin bone. On day 49, all calves were humanely euthanized at the MSU Meat Laboratory, and the right and left forelimbs were collected at the middle of the radius. From the right and left forelimb of each calf, synovial fluid was collected from the radiocarpal, middle carpal, lateral fetlock, and medial fetlock joints. Synovial fluid was kept on dry ice during sampling, after which it was placed in a −80 °C freezer until analysis. Cartilage from the proximal surface of the fused third and fourth metacarpal (MC III and IV) was collected from the right and left forelimb of each calf. Cartilage was kept on dry ice during sampling, then stored in a −20 °C freezer until analysis. Synovial fluid and cartilage samples were collected within 30 min of animal death. After cartilage and synovial fluid sample collection, the front legs of each calf were labeled and placed in a chiller (4.8 °C) overnight until computed tomography (CT) scanning.

### 2.3. Computed Tomography Scans

Computed tomography scans were performed at the MSU College of Veterinary Medicine in a GE Revolution Evo Scanner (Boston, MA, USA). Both the right and left forelimb of each calf were CT scanned within 36 h of euthanasia. The position was set to “lumbar spine”, slice thickness was set at 0.625-mm, with settings of 120 kV and 320 mAmp. The field of view was 180 mm with a 512 × 512 matrix size, leading to a pixel size of 0.35 mm × 0.35 mm. The voxel volume was 0.077 mm^3^ (0.35 mm × 0.35 mm × 0.625 mm). Calcium hydroxyapatite phantoms (Image Analysis, Inc; Colombia, KY, USA) were included in each scan, with rows representing 0, 75, and 100 mg mineral/cm^3^ for BMD comparison. All CT scans were analyzed with Mimics 24.0 software (Materialise, Leuven, Belgium). For each MC III and IV and lateral and medial proximal phalanx bone, the proximal and distal end were recorded to calculate the midpoint of the bone. Measurements of BMD, area, and cortical widths were performed at the midpoint of each MC III and IV and lateral and medial proximal phalanx. Bone mineral density, cross sectional area, cortical widths, and cortical areas were determined with a mask threshold value of 400 Hounsfield Units (HU). The moment of inertia (MOI) was calculated from the MC III and IV diameters measured in Mimics with the calculation for a hollow ellipse described in the American Society of Agricultural and Biological Engineers (ASABE) standards [2,24,26]. The values for average BMD are reported as HU. Average HU values were recorded at each of the 3 concentrations of calcium along the length of the phantom at 10 locations for each scan. The average HU values for each concentration were then compared in a scatter plot to the known concentrations of the phantom (Appendix A). An equation from the regression line converted HU to mg mineral/cm^3^. This method of determining BMD from the CT scans has been previously utilized [2,27].

### 2.4. Biomechanical Testing

Before biomechanical testing, front legs were removed from the freezer (−20 °C) to thaw for 5 days at 4.8 °C in order to clean the MC III and IV. Soft tissue and the remaining bones were removed with a scalpel from the fused MC III and IV to allow tensile testing of only the MC III and IV. Cleaned MC III and IV were then stored in the freezer until tensile testing (−20 °C). Three days before tensile testing, MC III and IV were removed from the freezer to thaw at 4.8 °C. Fracture force of the MC III and IV was determined by four-point bending at room temperature with an electromechanical testing system (MTS Criterion, Model 43), equipped with a 60 kN load cell. Left and right MC III and IV for each calf were placed individually, with the palmar aspect of the bone facing upwards toward the force applicators [2]. The bottom supports were 82-mm apart, and upper supports, which applied the vertical force from the load cell, were 52-mm apart. All samples were loaded to failure at a rate of 10 mm/min. Fracture force was recorded as the maximum force (ultimate load) before failure. Data acquisition rate was set to 100 Hz.

### 2.5. Osteocalcin

Calf serum samples were analyzed for OC concentration, a marker of bone formation reflecting osteoblastic activity, with the commercially available MicroVue Osteocalcin Enzyme Immunoassay (Quidel, San Diego, CA, USA). Calf serum samples utilized for OC analysis were diluted at a 1:15 ratio with a wash buffer for the samples taken through week 4. At week 4, depending on individual calf samples, sera were diluted at 1:20 or 1:25 with a wash buffer, as without this dilution, the samples were outside of the threshold of sensitivity. An analysis was performed according to the kit instructions. Coefficients of variation below 10% were accepted.

### 2.6. C-Telopeptide Crosslaps of Type I Collagen

Calf serum samples were analyzed for CTX-1 concentration, a marker of bone resorption, with the commercially available Serum CrossLaps kit made by Immunodiagnostics Systems (Gaithersburg, MD, USA). The samples were run neat and an analysis was performed according to the kit instructions. Coefficients of variation below 10% were accepted.

### 2.7. Procollagen II C-Propeptide

An ELISA kit from IBEX Pharmaceuticals (Montreal, QC, Canada) was obtained for analysis of CPII concentration in the serum, a marker of collagen synthesis. This competitive assay measures CPII, which is released from type-2 collagen during collagen synthesis. Serum samples were run in triplicate and with a 1:4 dilution, except for day 0, which was run with a 1:2 dilution. The assay procedure was performed according to instructions accompanying the kit. Coefficients of variation below 10% were accepted.

### 2.8. Prostaglandin E_2_ and Nitric Oxide

Calf synovial fluid samples were analyzed for concentration of PGE_2_ and NO, markers of inflammation. Fetlock joint synovial fluid samples were analyzed with the commercially available Thermo Fisher (Waltham, MA, USA) PGE_2_ ELISA kit. Due to supply chain issues, sufficient kits from Thermo Fisher for carpal synovial fluid were not able to be procured. Carpal synovial fluid samples were analyzed with a PGE_2_ ELISA kit made by Enzo (Farmingdale, NY, USA). Synovial fluid samples were digested with 50 µg/mL of hyaluronidase from bovine testes. Digested samples were diluted at a 1:2 ratio with reagent diluent and an analysis was performed according to kit instructions. Coefficients of variation below 10% were accepted.

Nitric oxide was measured by quantifying nitrite (an end product of nitric oxide metabolism) using a Greiss reaction and sodium nitrite standard [28,29]. The samples were not digested with hyaluronidase or diluted, as this caused sample readings to be too low and outside of the threshold of sensitivity. Nitric oxide results are expressed in micromoles per well. Coefficients of variation below 30%, larger than with other assays, were accepted due to the variability in undigested and undiluted synovial fluid.

### 2.9. Glycosaminoglycan

Cartilage slices from the proximal surface of the MC III and IV were digested with papain to determine the GAG concentration with a dimethylmethylene blue assay [30]. This assay is based on the binding of anionic GAGs to cationic 1,9-dimethylmethylene blue. Sulfated GAG content was measured against a chondroitin sulfate standard, and the sample concentration was determined with a linear curve [28]. Papain digested samples were diluted at a 1:25 with a sodium acetate and tween dilution buffer. Coefficients of variation below 10% were accepted.

### 2.10. Statistical Analysis

All statistical analyses were performed in SAS 9.4. The residuals were plotted against the predicted means and observed for normality. Nitric oxide data were found to be abnormally distributed and were log transformed to achieve normal distribution. All other data were found to be normally distributed. Height, weight, length, OC, CTX-1, and CPII were evaluated with the repeated effect of day, with calf as the subject, fixed effects of day and treatment, as well as the interaction between day and treatment. Repeated measurements of OC, CTX-1, and CPII were run with d 0 as a covariate. Fracture force and glycosaminoglycan content were evaluated with fixed effects of leg (right or left) and treatment, as well as the interaction between leg and treatment, with calf as a random effect. Fetlock NO and PGE_2_ concentration were evaluated with the fixed effects of treatment and joint (lateral or medial fetlock) and the interaction between treatment and joint. Carpal NO and PGE_2_ concentration were evaluated with fixed effects of treatment and joint (radiocarpal or middle carpal) and the interaction between treatment and joint. Calf was included as a random effect for both fetlock and carpal NO and PGE_2_. Fused MC III and IV CT data were evaluated with the fixed effects of treatment and leg, as well as the interaction between treatment and leg and calf was included as a random effect. Proximal phalanx CT data were evaluated with the fixed effects of treatment, leg, and bone (lateral or medial phalanx) and interactions and calf was included as a random effect. Means are reported as averages ± standard error of the mean (SEM). Error bars in graphs represent SEM. Statistical significance was set at *p* ≤ 0.05 and trends at *p* ≤ 0.10.

## 3. Results

Study day had a significant effect on calf height (*p* < 0.001), weight (*p* < 0.001), and length (*p* < 0.001), supporting animal growth throughout the study (Appendix A). The interaction between day and treatment was not significant for calf height, weight, or length (*p* = 0.99, *p* = 0.99, and *p* = 0.68, respectively). Average daily gain (ADG) was not different among the treatments (Table 2). On both d 0 and d 48, calf height, weight, and length were not different among the treatments (Table 2).

### 3.1. Fused MC III and IV CT Scan Results

There were no differences in MC III and IV internal or external diameters at the midpoint of the bone between left and right leg or treatments. There was no interaction between treatment and leg for dorsopalmar internal diameter (*p* = 0.54), mediolateral internal diameter (*p* = 0.42), or mediolateral external diameter (*p* = 0.54). There was a trend for an interaction between treatment and leg for dorsopalmar external diameter (*p* = 0.09). This interaction was driven by the right leg of the small circle treatment group having a larger dorsopalmar external diameter than the left leg (*p* = 0.04). The moment of inertia was not different among treatments (*p* = 0.50, Appendix A), but there was a trend for an effect of leg (*p* = 0.10), with the left leg having a higher moment of inertia. There was no interaction between treatment and leg (*p* = 0.15).

The bone length of MC III and IV was different among treatments (Table 3, *p* = 0.04), with the large circle treatment group having shorter MC III and IV compared to all other treatments. There was no difference between left and right leg length (0.57) and no interaction between treatment and leg (*p* = 0.96). The cortical area was not different between leg (*p* = 0.15), or treatment (*p* = 0.057), but there was a trend for an interaction between treatment and leg (*p* = 0.07). This trend was driven by the left leg MC III and IV of the treadmill group having a larger cortical area then the right leg (*p* = 0.01). The cross-sectional area was not different between left and right legs (*p* = 0.42), treatment (*p* = 0.026), and there was no interaction between treatment and leg (*p* = 0.36). 

For the dorsal, lateral, medial, and palmar cortices, as well as the entire slice of the midpoint, BMD was not different between left and right legs (*p* = 0.73, 0.64, 0.17, 0.82, and 0.12, respectively) nor treatment (0.95, 0.73, 0.76, 0.97, and 0.96, respectively, Appendix A). No interactions between leg and treatment were significant.

Dorsal, lateral, medial, and palmar cortical widths of the MC III and IV were not different among treatments (0.38, 0.86, 0.63, and 0.70, respectively, Appendix A). The medial cortex width of the MC III and IV tended to be larger for the left leg compared to the right leg (*p* = 0.063, Appendix A). No interactions between leg and treatment were significant.

### 3.2. Medial and Lateral Proximal Phalanx CT Scan Results

Dorsopalmar internal and external diameters were not different among treatments (*p* = 0.68, 0.70), leg (*p* = 0.85, 0.64), or bone (*p* = 0.36, 0.53). Mediolateral internal and external diameters were not different among treatments (*p* = 0.90, 0.56) or leg (*p* = 0.84, 0.61). There was a difference between the proximal phalanx bones, with the medial proximal phalanx having a greater mediolateral internal (*p* = 0.0003) and mediolateral external diameter (*p* = 0.01). The interaction between treatment and bone was significant for the mediolateral external diameter (Table 4, *p* = 0.01).

Proximal phalanx bone length was not different among treatments (*p* = 0.40), lateral or medial bones (*p* = 0.76), or the left and right legs (*p* = 0.72). No interactions were significant. The cortical area was not different among treatments (*p* = 0.65) or legs (*p* = 0.17), but was different between the lateral and medial proximal phalanx bones (*p* = 0.01), with the medial phalanx having a greater cortical area. No interactions were significant. The cross-sectional area was not different among treatments (*p* = 0.71), leg (*p* = 0.83) or proximal phalanx bones (*p* = 0.18). There was a tendency for an interaction between leg and bone (*p* = 0.07). This trend was driven by the right medial proximal phalanx having a larger area than the right lateral proximal phalanx (*p* = 0.03).

For the dorsal cortex, BMD was not different among treatments (*p* = 0.59) or leg (*p* = 0.79), but there was a trend for an interaction between leg and bone (Table 5, *p* = 0.07). Lateral cortical BMD was not different among treatments (*p* = 0.69) or leg (*p* = 0.75) but was different between the proximal phalanx bones (*p* < 0.0001), with the lateral proximal phalanx having a greater lateral cortical BMD than the medial proximal phalanx. Medial cortical BMD was not different among treatments (*p* = 0.30) or leg (*p* = 0.66), but was different between the proximal phalanx bones (*p* < 0.0001), with the medial proximal phalanx having a greater cortical BMD. There was a trend for an interaction between leg and bone (Table 5, *p* = 0.06). For the palmar cortex, BMD was not different among treatments, legs, or proximal phalanx bones (*p* = 0.75, *p* = 0.34 and *p* = 0.20, respectively). There was a significant interaction between leg and bone (Table 5, *p* = 0.03) and a trend for an interaction between the treatment and proximal phalanx bone (*p* = 0.07). This trend was driven by the palmar cortex in the medial proximal phalanxes of the treadmill group having greater density than the palmar cortex of the lateral proximal phalanxes (*p* = 0.005). For midpoint slice BMD, treatments and legs were not different (*p* = 0.67 and *p* = 0.40, respectively). There was a tendency for a difference between the proximal phalanx bones (*p* = 0.08), with the medial proximal phalanx trending towards greater BMD. There was a significant interaction between leg and bone (Table 5, *p* = 0.001).

Dorsal, lateral, medial, and palmar cortical widths all had significant effects on the phalanx bone (Table 6). There were no effects for treatments or legs on the cortical widths of the dorsal, lateral, medial, or palmar cortices.

### 3.3. Fracture Force of MC III and IV

There was no significant effect of the treatment (*p* = 0.70) or leg (*p* = 0.88), but there was an interaction between treatment and leg (Table 7, *p* = 0.05), with the right fused MC III and IV of the treadmill treatment having a lower fracture force than the left.

### 3.4. Average Serum OC

There was a significant effect of day and treatment on average OC concentration (Figure 1, *p* < 0.001 and *p* = 0.049, respectively), with the small circle and treadmill exercise groups having a greater osteocalcin concentration than the control group. Overall osteocalcin concentrations were higher on d 14 and 21. The interaction between day and treatment was not significant (*p* = 0.97).

### 3.5. Average Serum CTX-1

There was a significant effect of day on serum CTX-1 concentration (Figure 2, *p* = 0.02) but not treatment (*p* = 0.15). Day 0 had a greater average concentration of CTX-1 than other dates. The interaction between day and treatment was not significant (*p* = 0.42).

### 3.6. Average Serum CPII

There was a trend for a treatment effect on serum CPII concentration (*p* = 0.08) and a day effect (*p* < 0.0001), with day 14, 21, 28, and 49 having lower concentrations than other days (Figure 3). The interaction between day and treatment was not significant (*p* = 0.73).

### 3.7. Fetlock Synovial Fluid NO

The treatments had a significant effect on synovial fluid NO concentration (Figure 4, *p* = 0.0005,) with the large circle treatment having the highest concentration of NO. Joints did not have a significant effect (*p* = 0.80), nor was the interaction of treatment and joint significant (*p* = 0.51).

### 3.8. Carpal Synovial Fluid NO

The treatments did not have a significant effect on synovial fluid NO concentration (*p* = 0.27), but joints did (Figure 5, *p* < 0.0001), with middle carpal joints having a greater average NO concentration than radiocarpal joints. The interaction of treatment and joint was not significant (*p* = 0.91).

### 3.9. Fetlock Synovial Fluid PGE_2_

The treatments did not have a significant effect on synovial fluid PGE_2_ (*p* = 0.99) but joints did (Figure 6, *p* = 0.0005), with both medial fetlock joints having a greater average PGE_2_ concentration than the left lateral fetlock joint. The interaction of treatment and joint was not significant (*p* = 0.16).

### 3.10. Carpal Synovial Fluid PGE_2_

The treatments did not have a significant effect on synovial fluid PGE_2_ (*p* = 0.78) but joints did (Figure 7, *p* = 0.007), with the right radiocarpal joint having a higher average concentration than both left and right middle carpal joints. The interaction of treatment and joint was not significant (*p* = 0.60).

### 3.11. Cartilage Glyconaminoglycan Content

There was no significant effect of the treatment (*p* = 0.73) or leg (*p* = 0.35), nor was the interaction between treatment and leg significant (*p* = 0.14). Within the control, large circle, and treadmill treatments, the left and right legs did not have different GAG concentrations at the proximal surface of the MC III and IV (Table 8). However, it was determined that the left leg of the small circle exercise treatment had a greater GAG concentration at the proximal surface of the MC III and IV than the right leg (Table 8, *p* = 0.03).

## 4. Discussion

This study utilized calves as a model for young horses to determine the impact that circle diameter can have to joint and bone health of the front limbs. It was hypothesized that a smaller diameter circle would lead to asymmetry between the inside and outside forelimbs and increased markers of joint inflammation and cartilage metabolism. This hypothesis is partially accepted, as bone morphology and cartilage GAG content differences were noted within the small circle exercise group, but differences in the markers of joint inflammation were not observed within the small circular exercise group.

Computed tomography scans in this study found the right (inside) leg of the small circle treatment group to have a larger dorsopalmar external diameter than the left (outside) leg. In racing Thoroughbreds, the inside forelimb is the most common location for injuries (56% of catastrophic musculoskeletal injuries) [31]. While galloping around a turn, the outside front limb has a pushing function, and the weight of the horse will be pushed forward and laterally onto the lead forelimb [32]. While there is a great speed difference between the walking gait of calves in this study and Thoroughbreds galloping around a turn, the location of bone adaptation in the small circle exercise calves seems to be a result of exercising on a turn.

Computed tomography analysis also identified treatment differences in the more-distally located proximal phalanxes. The small circle exercise group was found to have a greater external mediolateral diameter in the medial phalanxes compared to the lateral phalanxes. This was also noted in the control treatment group. Both treadmill and large circle exercised animals had similar-sized lateral and medial phalanx external mediolateral diameters. The similarity of the straight-line treadmill exercise and large circle exercise diameters between the two phalanxes may suggest that the large circle exercise is more similar to straight-line exercise than the small circle exercise at such low speeds. Calves in the control treatment only had access to movement in the group housing pen. These movements may have included tight turns similar to the small circle exercise group.

Cortical density and widths of the proximal phalanxes were not impacted by treatment, but were, however, different between the lateral and medial proximal phalanx bones. Very few studies have evaluated cortical bone characteristics of the proximal phalanxes in exercising horses or bovines. Horses have a single proximal phalanx unlike calves. This current study provides the necessary categorization of the differences in bovine lateral and medial proximal phalanxes, which is important in the continued use of cloven-hoof species, such as calves, to serve as a skeletal model for equines. The lack of treatment differences between cortical widths and BMD from computed tomography are not surprising. It is known that dynamic exercise, such as sprinting, leads to bone adaptations [2,24,25,33]. As has been previously determined, gait is an important factor in the response of loading during circular exercise [11]. In this study, exercise only occurred at slow speeds of 1.1–1.5 m/s, leading to a walking gait for the calves, and no treatment differences between cortical widths or BMD. Based on the mechanostat theorem, localized strain within 1500–2500 microstrain will lead to bone maintenance, but greater strain will lead to a formation response in bone in order to reduce strain [1,34]. A limitation of this current study is that strain was not evaluated or calculated. Stride rate or other gait characteristics were not recorded in this study to allow for strain calculation. Strain evaluation among the cortexes of the MC III and IV would aid in the determination of calf bone response to circular exercise, treadmill exercise, as well as the free exercise in group housing.

The fracture force of the MC III and IV was found to be lower for the right leg of the treadmill exercise treatment group compared to the left. Similarly, the left MC III and IV of the treadmill group was found to have a larger cortical area than the right leg. During treadmill exercise, the calves wore rope halters, which were tied with a quick release knot to either the left or the right side of the treadmill, depending on which side of the treadmill the calves were on. Depending on calf size, between two and four calves could fit on the treadmill at a time. During the last week of the study, the calves were large enough that only two calves could fit on the treadmill at once, both tied to the left side of the treadmill as the handlers were only able to stand on the left side of the treadmill. While the calves were tied with their rope halter loose enough so that they were encouraged to travel straight, there is a potential that having their rope halter tied to their left side could have caused asymmetric loading between the right and left forelimb. Kinematics between treadmill and over ground exercise are not identical. Exercise on a treadmill exposes animals to footing that is different than that of normal ground surfaces. Treadmill exercise can impact gait factors, such as a longer stance duration of the forelimbs at the trot, less lumbar motion at the trot, and longer stride length at the canter [35]. The difference between the right and left MC III and IV fracture force may have simply been a result of a small number of animals on each treatment, and not method of exercise, but both possibilities should be noted. The presence of a shorter MC III and IV for the large circular exercise group is most likely due to the very little variation in bone length within and among treatments, causing for a relatively small SEM that could allow for any minor difference, whether related to exercise treatment or not, to be found significant.

Similar to bone morphology, some treatment differences, as well as multiple joint differences, were found in the markers of joint inflammation and metabolism, NO and PGE_2_. Average fetlock NO concentration was found to be greatest in the large circular exercise treatment. During the first two weeks of exercise, the large circle calves were reluctant to exercise, and while they maintained an average speed within the range of this study, they often paused during exercise and needed verbal encouragement to begin walking again. Once they began walking again, they needed to take a few steps at a faster stride to avoid getting shocked by the panels in the electric walker. The small circle exercise and treadmill groups exercised continually with no pauses in a walking session, unlike the large circle group. This behavioral difference may contribute to the greater concentration of NO in the large circle exercise groups. It is also worth noting that in this study, there was large variation in NO concentration in the synovial fluid samples due to the viscous nature of un-diluted and un-digested synovial fluid. Future studies may have more success evaluating the NO concentration in serum.

In this study, we specifically chose not to compare the results of the fetlock synovial fluid to the carpal synovial fluid, understanding that the range of motion and relative loading of these joints are vastly different due to their locations. However, differences within the fetlock and carpal joints were found. The carpal NO concentration was different between locations, with middle carpal joints having higher NO concentrations than radiocarpal joints, providing further categorization of the joint characteristics of cloven-hooved animals. Nitric oxide is considered to be important in the initiation of repair and attracts bone cells to the site of injury. NO may also be involved in the regulation of osteoclasts, as in vitro osteoclast cell death has been found after high doses of NO [36]. NO can lead to the dysregulation of osteoblast and osteoclast balance, which can result in cartilage destruction through chondrocyte apoptosis [37]. Fetlock PGE_2_ was greater in medial fetlock joints than lateral fetlock joints of both legs. Carpal PGE_2_ was lower for the right middle carpal compared to left and right radiocarpal joints, regardless of treatment. Synovial fluid PGE_2_ in two-year-old horses and dogs has been found to increase after surgically-induced osteoarthritis [19,38]. Middle carpal NO concentration was greater than radiocarpal, but middle carpal PGE_2_ concentration was lower than the radiocarpal. Injuries to carpal joints have been found to vary based on animal function. Middle carpal joint injuries are found frequently in racing horses due to repeated trauma, but pleasure and sport horses are found with osteoarthritis most commonly in the radiocarpal joint [39]. Exercise performed in this study did not impact fetlock or carpal PGE_2_ concentration. In this study, synovial fluid concentrations of NO and PGE_2_ were mostly influenced by joint location, similar to the proximal phalanx bone location. These results provide important information on joint-based differences of the biomarkers in calves, which have not been previously analyzed as a result of exercise.

The determination of GAG content in the proximal surface of MC III and IV was not different among treatments. It has been previously speculated that in 15-week-old calves, the joints may still be homogeneous and GAG content was not influenced by short-duration exercise [23]. In this study, within the small circle exercise group, the left (outside) leg had greater GAG content in the MC III and IV proximal surface compared to the right (inside) leg. Another study found that, after a 5-month period, horses confined to box stalls had elevated cartilage GAG content compared to horses that were afforded pasture access [40]. As has been previously discussed, the lean angle increases as a result of smaller radii circles and has the potential for uneven loading between the inside and outside limbs [6,14,41]. With only six animals per treatment, and exercise at a slow rate of speed in this study, treatment differences may not have been detectable. However, a future experiment with a combination of more animals and higher speeds may find GAG content to be different between the cartilage surfaces of the front limbs.

Average OC, a marker of osteoblastic activity and, thus, bone formation [42], was greater for the small circle and treadmill exercise treatments compared to the control treatment. However, the large circle group was not greater than the control exercise group. During the first two weeks of exercise, the large circular exercise calves were reluctant to exercise and needed encouragement to achieve the basic requirements of exercise at 1.1–1.5 m/s. However, the treadmill and small circle exercise treatment groups were amenable to exercise and needed less auditory coercion to continue to exercise at the speed required for the study. In this study, behavior was not evaluated or analyzed, but the described behavioral differences may have been a contributing factor to the treatment differences in average OC concentration. A limitation of this study is the lack of internal load indicators evaluated during exercise. Future studies including an analysis of internal load of exercising calves, such as heart rate response, can aid in characterizing more of the psychophysiological response of calves to exercise [43].

Average OC, CTX-1, and CPII were different among treatments on d 0, thus, d 0 was a significant covariate for all serum markers. All the calves were randomly assigned to treatments and striated based on weight at d 0, assuring that each treatment had calves of equal sizes at the start of the study. The calves were transported to the farm 1 week before day 0. In future studies utilizing calves for exercise, animals may need more than one week for housing acclimation before beginning exercise. It is interesting to note that in the middle of the study, on days 14, 21, and 28, CPII concentration appears to decrease, then increase after day 28. At this time, OC was also elevated. By day 14, all treatment groups were acclimated to their exercise treatments and successfully exercising with little coercion. These day differences observed may be a result of calves settling into their exercise and housing after acclimation. The lack of day effects in CTX-1 after day 0 is not surprising, as calf serum CTX-1 has not previously exhibited a day effect during sprint exercise or confinement [2].

## 5. Conclusions

Circular exercise is used frequently, and by varying methods across the equine industry. Circle size, speed of travel, and training of the animal are all factors that should be considered when utilizing circular exercise. Based on the results of this study, altering circle size can impact joint and bone health, with a smaller circle size leading to differences in bone diameters, as well as cartilage glycosaminoglycan content. This study provides the initial characterization of physiological responses to circular exercise performed by calves on a walker at slow speeds. The results from this study, coupled with other studies available in the literature, suggest that circular exercise, even at a slow speed, can impact joint and bone health of young animals. The circular exercise in this study was performed at slow speeds, and for a duration of 7 weeks, a very short time span in relation to horse training. The alteration of speed or duration of exercise could eventually lead to greater changes in bone morphology and biomarkers. Further information needs to be explored on circular exercise, such as the impact of different styles of riding, presence of a rider, and effect on the hind limb function. Handlers and riders utilizing circular exercise should recognize that the manner in which they exercise animals can impact overall health, and should consider the circle size at which animals exercise to be a factor contributing to bone and joint health.

## Figures and Tables

**Figure 1 animals-12-01379-f001:**
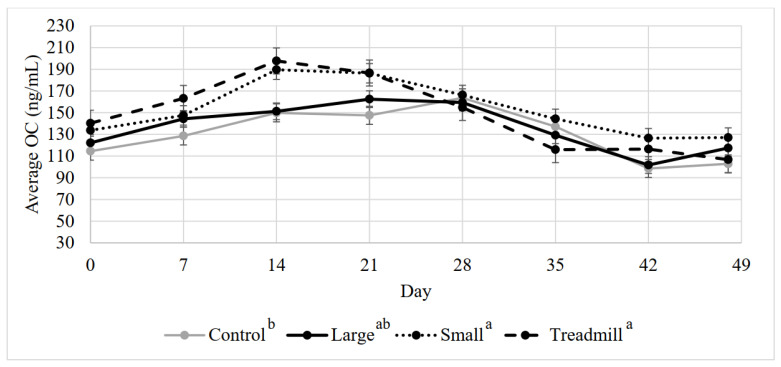
Mean calf serum osteocalcin (OC) by treatment throughout the 7-week study period. ^a, b^ Treatments lacking common superscripts differ (*p* = 0.05).

**Figure 2 animals-12-01379-f002:**
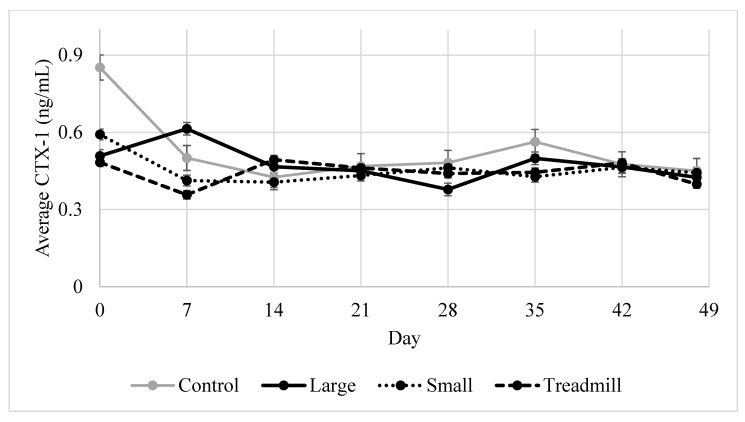
Mean calf serum C-telopeptide crosslaps of type I collagen (CTX-1) by treatment throughout the 7-week study period.

**Figure 3 animals-12-01379-f003:**
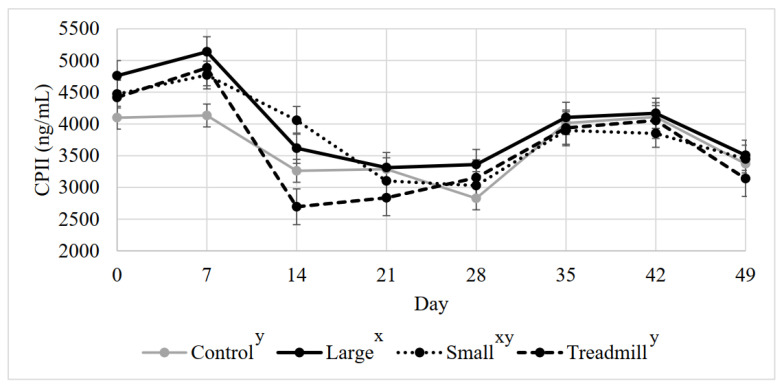
Mean calf procollagen II c-propeptide (CPII) by treatment throughout the 7-week study period. ^x, y^ Treatments lacking common superscripts have a tendency to differ (*p* = 0.08).

**Figure 4 animals-12-01379-f004:**
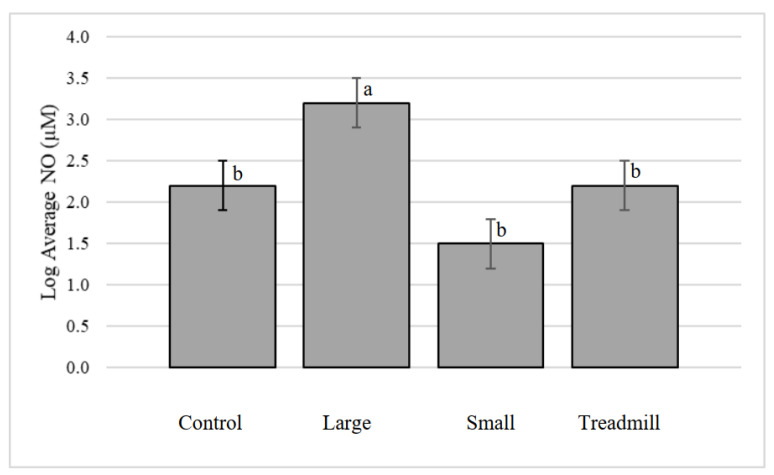
Log transformed average nitric oxide (NO) concentration in the lateral and medial fetlock joints by treatment. ^a, b^ Treatments that lack common superscripts differ (*p* = 0.0005).

**Figure 5 animals-12-01379-f005:**
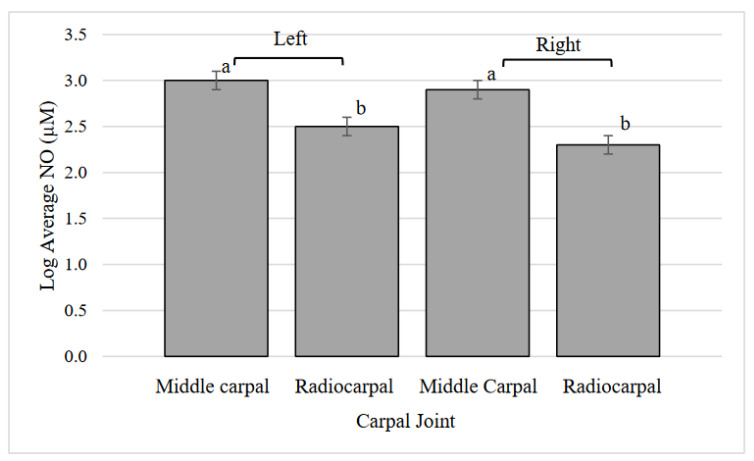
Log transformed average nitric oxide (NO) concentration in the middle carpal and radiocarpal joints. ^a, b^ Values that lack common superscripts differ (*p* < 0.0001).

**Figure 6 animals-12-01379-f006:**
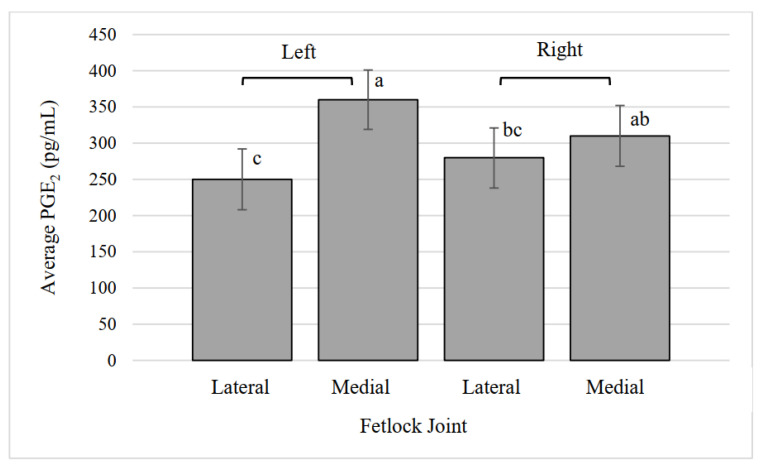
Average prostaglandin E_2_ (PGE_2_) concentration in lateral and medial fetlock joints. ^a,b,c^ Values that lack common superscripts differ (*p* = 0.0004).

**Figure 7 animals-12-01379-f007:**
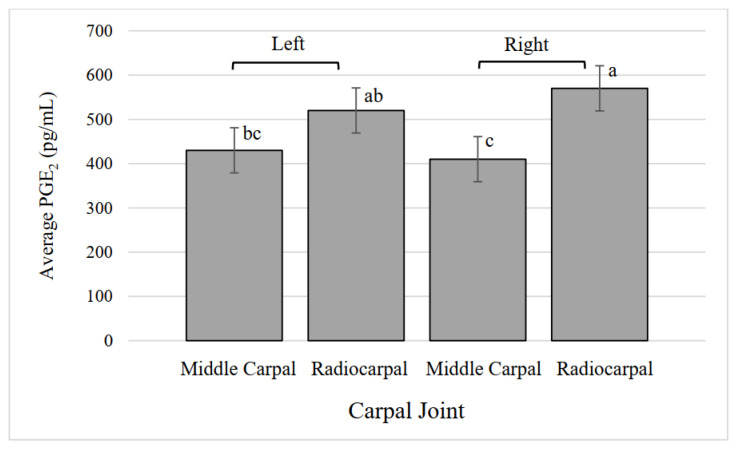
Average prostaglandin E_2_ (PGE_2_) concentration in middle carpal and radiocarpal joints. ^a,b,c^ Values that lack common superscripts differ (*p* = 0.007).

**Table 1 animals-12-01379-t001:** Exercise protocol for treadmill, small circle, and large circle treatment groups. All exercise was performed 5 d/wk at a speed of 1.1–1.5 m/s.

Week	Exercise Duration
1	5 min/d
2	10 min/d
3	15 min/d
4	20 min/d
5	25 min/d
6	30 min/d
7	30 min/d

**Table 2 animals-12-01379-t002:** Initial (d 0) and final (d 48) calf height, weight, and length by treatment, as well as average daily gain (ADG).

Treatment	Height (d 0) cm	Height (d 48) cm	Weight (d 0) kg	Weight (d 48) kg	Length (d 0) cm	Length (d 48) cm	ADG, kg
Control	89	100	75	146	90	113	1.5
Large	88	100	81	145	90	112	1.3
Small	89	102	81	156	89	113	1.6
Treadmill	87	100	80	145	87	114	1.4
SEM	1.2	1.1	4.0	6.0	2.02	1.2	0.1
*n*	6	6	6	6	6	6	6
*p*-Value	0.52	0.51	0.71	0.44	0.71	0.61	0.17

**Table 3 animals-12-01379-t003:** Bone length of the right and left metacarpal III and IV (MC III and IV) by treatment.

Treatment	MC III and IV Bone Length, mm
Control	199 ^a^
Large	192 ^b^
Small	195 ^a^
Treadmill	195 ^a^
*n*	12
SEM	1.6
*p*-Value	0.04

^a, b^ Values that lack common superscripts within a column differ (*p* = 0.04).

**Table 4 animals-12-01379-t004:** Internal (int) and external (ext) cortical diameters of the lateral and medial proximal phalanx of both front legs.

Treatment	Proximal Phalanx Bone	Dorsopalmar Int Diameter, mm	Dorsopalmar Ext Diameter, mm	Mediolateral Int Dimeter, mm	Mediolateral Ext Diameter, mm
Control	Lateral	20.7	25.5	17.2	23.6 *
	Medial	20.4	25.4	18.0	24.5 *
Large	Lateral	19.8	25.0	17.2	23.6
	Medial	20.1	25.2	17.4	23.8
Small	Lateral	20.1	25.4	17.1	23.7 *
	Medial	20.6	25.7	18.0	24.8 *
Treadmill	Lateral	19.9	24.9	17.2	23.5
	Medial	20.0	24.9	17.4	23.6
*n*		12	12	12	12
SEM		0.45	0.42	0.42	0.41
*p*-Value		0.44	0.63	0.26	0.01

* Denotes lateral and medial proximal phalanx are different within a treatment (*p* = 0.01).

**Table 5 animals-12-01379-t005:** Cortical bone mineral density of the lateral and medial proximal phalanx of left and right legs.

Leg	Proximal Phalanx Bone	Dorsal Cortex, mg Mineral/cm^3^	Lateral Cortex, mg Mineral/cm^3^	Medial Cortex, mg Mineral/cm^3^	Palmar Cortex, mg Mineral/cm^3^	Midpoint Slice, mg Mineral/cm^3^
Left	Lateral	845	959	728 *	799 *	737 *
	Medial	855	776	974 *	837 *	764 *
Right	Lateral	853 ^#^	973	769 *	812	759
	Medial	831 ^#^	755	949 *	803	751
*n*		24	24	24	24	24
SEM		16	21	23	17	12
*p*-Value		0.07	0.19	0.06	0.03	0.001

* Denotes lateral and medial proximal phalanx are different within the left or right leg (*p* < 0.05). ^#^ Denotes lateral and medial proximal phalanx tend to differ within the left or right leg (*p* < 0.10).

**Table 6 animals-12-01379-t006:** Cortical widths of the lateral and medial proximal phalanx of left and right front limbs.

Proximal Phalanx Bone	Dorsal Cortex, mm	Lateral Cortex, mm	Medial Cortex, mm	Palmar Cortex, mm
Lateral	2.6 ^a^	3.7 ^a^	2.6 ^b^	2.6 ^a^
Medial	2.5 ^b^	2.6 ^b^	3.8 ^a^	2.7 ^b^
*n*	48	48	48	48
SEM	0.07	0.12	0.11	0.06
*p*-Value	0.04	<0.0001	<0.0001	0.02

^a, b^ Values that lack common superscripts within a column differ (*p* < 0.05).

**Table 7 animals-12-01379-t007:** Fracture force of calf MC III and IV by treatment and leg.

Treatment	Leg	Force (N)
Control	Left	8700
	Right	9300
Large	Left	8800
	Right	9200
Small	Left	10,100
	Right	10,400
Treadmill	Left	9400 *
	Right	8300 *
*n*		6
SEM		960
*p*-Value		0.05

* Denotes lateral and medial proximal phalanx are different within the left or right leg (*p* < 0.05).

**Table 8 animals-12-01379-t008:** Average glycosaminoglycan concentration in cartilage slices from the proximal surface of the MC III and IV by treatment and leg.

Treatment	Leg	GAG (mg/g)
Control	Left	88
	Right	90
Large	Left	84
	Right	71
Small	Left	113 ^a^
	Right	68 ^b^
Treadmill	Left	79
	Right	99
*n*		6
SEM		14
*p*-Value		0.14

^a,b^ Values that lack common superscripts differ (*p* < 0.05).

## Data Availability

Datasets used and analyzed during the present study are included in the article and Appendix A. Raw data and further inquiries can be directed to the corresponding author.

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
