# Peer review of "The Impact of Circular Exercise Diameter on Bone and Joint Health of Juvenile Animals"

_animals, 2022, doi:10.3390/ani12111379_

Round 1

Reviewer 1 Report

well-written manuscript materials and methods are efficient and sn results in line with research These differences suggest that altering the circular diameter of exercise can affect bone health and procedures and that larger diameters can prevent asymmetrical loading between the inner and outer legs.

Author Response

The authors appreciate the time this reviewer took to evaluate our publication. We are grateful our publication has been found to be appropriate for this journal, with well-presented results.

Reviewer 2 Report

General comments

The study is interesting, particularly given the low expected microstrain on the respective bones with the walking gait.  The lack of framing of bone response, and even the biomarkers, back to the respective expected micro strain is a limitation with the manuscript that needs to be addressed. 

Specific comments

Please check formatting of authors names  eg ……………..and Jr

Introduction

Given that the authors are attempting to use the walking gait in calves as a model for horses galloping there is a need to frame the introduction in relation to the mechanostat and the respective microstrain experienced by the phalanges and mc3 (mc3/mc4) during locomotion and during turning.  The lack of reference back to the mechanostat theorem this is a major limitation of the introduction and the discussion.

Even if there is little information on calves or bovine the information on horses provides a framework to demonstrate the relative increases in micro strain associated with increasing velocity.  The use of walking gait – while understandable from a practical point of view does represent relatively low potential loading.

Line 68 please check this reference – Parkes et al did not measure or calculate GRF  perhaps you meant to reference chateau et al 2013

Line 74 it needs to be clear that this sentence relating to ref 8 is for humans not horses

Lines 77-83 could have better development, please look at the Parkes paper  you had as ref 5 – there is a need to bring this back to magnitude and location of microstrain

Line 85 – not sure what the relevance is of the  jockey injury paper to this study  – please delete this sentence as it does not fit with paragraph

Line 114 there is a need for a more precision with the referencing – the sentence refers to calves – but ref 25 is for horses  - please either correct sentence or change reference

Line 120 operation is probably appropriate for a USA audience – consider changing to farm for international audience

Line 124 – calf starter – assume this is a pelleted ration – if yes provide manufacturer etc details

Materials and methods

Line 149  consider changing scale to weight scale or similar

Section 2.3 computed tomography

Line 165 please specify voxel size

Line 174 – should be moment of inertia

Lines 184-190 please consider rewriting or editing this – in its current form the reader has to re-read this to understand why you mention two freeze thaw cycles.  The use of individual mc3 & 4 and then in another sentence fused mc3 & 4 could cause the reader confusion – consider alternative sentence structure 

Line 193 and line 195  consider deleting mention of loading speed in line 193 as this is essentially repeated in line 195

Results

Please provide summary values for the ADG within the text

Line 346 please use correct anatomical terms – what is a fetlock bone?

Line 354 …lower fracture force….

Figures 1, 2 3

Please provide within the specific timepoints the periods when treatments differed

Discussion

The discussion is fair but really needs to be brought back to the primary aspect of bone response which is appropriate number of cycles above 2000 microstrain

You can reverse calculate the estimated number of load cycles (strides) and the magnitude of strain from the published data and gait speed

Carpal joint NO and PGE2 – you have a joint location effect here – so this needs to be described in relation to the relative loading (and range of motion)  in the different joints

Reviewer 3 Report

With interest, I have been reading the manuscript entitled: "The impact of circular exercise diameter on bone and joint health of juvenile animals " by Logan et al.

Congratulations on the manuscript. I have imagined how much hard work was involved, and I suppose it may have contributed to other researchers' training within the research group. Overall, article will contribute to understanding the joint and bone development in the calves is worthy of publication. The work is written in understandable language. My main concern with this manuscript is the lack of an internal load description performed by calves in the methodology and results. I have a few comments and questions that I have detailed below.

Suggestions for improving the manuscript:

Simple summary/Abstract:

My comments: Overall, I feel that the review question is not addressed adequately in the abstract/simple summary onset.

I would like to see a brief rationale for monitoring exercise in calves is essential for the equestrian discipline. Moreover, the abstract does not contain any statistical analysis data or indication.

Introduction

My comments: Well written and straightforward. 

Materials and methods

L135-136: “5 d/wk starting at 5 135 min/d and increased by 5 min each week until reaching 30 min” and L137: “exercise was performed at a speed of 1.1–1.5 m/s”

The above  information is very general and may induce misinterpretation. The authors should include the lack of internal load information as a “study limitation” in the discussion.

The concept of internal load consolidates all the psychophysiological feedback occurring during the execution of the exercise, exemplifying by increases in heart rate or blood lactate concentration means the internal load produced by the external load (distance covered and duration). The internal load may reflect the psychophysiological response that the body’s calves initiate to cope with the requirements elicited by the external load. In the case of calves used herein, the external load has been represented for treadmill, small circle, and large circle performed at a speed of 1.1–1.5 m/s.

For more information, see: https://doi.org/10.1123/ijspp.2018-0935 and apply it in the discussion. 

The internal load could facilitate the comparison of the exercise loads between other studies on bone and joint health of juvenile animals regardless of the exercise surface. 

Round 2

Reviewer 2 Report

Thank you for your edits to the manuscript.  The changes are acceptable but did leave one felling the changes were pitched at the minimum alteration and development required. I was hoping the requests would provide a more robust development of the introduction and discussion.  The manuscript really still requires  exploration of the biological basis of the results observed based on the anatomical location of the response and the strain experienced.  It is very interesting that what theoretically should be a low magnitude of strain experienced during the walk was sufficient to stimulate the response observed.